# Optimization of water and land allocation in salinity and deficit- irrigation conditions at farm level in Qazvin plain

Sara Bulukazari[1], Hossein Babazadeh[1]*, Niazali Ebrahimipak[2], Seyed-Habib Mousavi-Jahromi[3], Hadi Ramezani Etedali[4]

1 Department of Water Science and Engineering, Science and Research Branch, Islamic Azad University, Tehran, Iran, 2 Department of Irrigation, Soil and Water Research Institute, Agricultural Research, Education and Promotion Organization, Karaj, Iran, 3 Department of Civil Engineering, Shahr-e-Qods Branch, Islamic Azad University, Tehran, Iran, 4 Department of Water Sciences and Engineering, Imam Khomeini International University, Qazvin, Iran

* h_babazadeh@hotmail.com

**Data Availability Statement:** All relevant data is within the paper and Supporting information files.

**Funding:** The author(s) received no specific funding for this work.

## Abstract

Improper extraction of water from resources especially in arid and semi-arid regions leads to a decrease in the quality of water and soil resources. In such areas, management activities such as increasing water productivity in agricultural sector would be a key step towards sustainable development. Therefore, water resources management to improve the allocation of limited water supplies is essential. In this study, a non-linear programming optimization model have been combined with a AquaCrop model to determine the optimal water and land allocation considering the quality issues of both water and soil resources with focusing on enhancing agriculture water productivity. For this purpose, the spatial variations of chemical and physical properties of soil in the Qazvin plain were taken into account. The soil of study site was divided into three salinity classes, and three weather conditions were identified by Standardized Precipitation Index (SPI). Moreover, five irrigation strategies were modeled under each weather condition. To understand the response of major crops under cultivation to water and salinity, the AquaCrop model was calibrated and validated (2005–2020) and utilized in the objective function. Accordingly, the production functions of the different products were obtained, and the cultivation area as well as amount of water consumption of the crops were optimized by using the target functions of maximum net income and maximum water use efficiency. The results showed that the model is capable of simulating crop yield in salinity and water deficit conditions. The coefficient of determination ($R^2$) for barley, wheat and maize was equal to 0.86, 0.92, and 0.96, respectively. Findings reveal that total irrigation water could be reduced by 20% on average without profit reduction when compared to the profit of the present situation. Total economic profit could be increased by 18% on average through the optimization of water allocation and cropping pattern with the same water supply amount as that of the current situation. Also, the water productivity increased between 12 to 30% under these conditions. Therefore, the proposed model can efficiently optimize the amount of irrigation water and cultivation area on a regional scale considering salinity conditions.

**Competing interests:** The authors have declared that no competing interests exist.

## Introduction

Iran has always been facing water shortage due to its arid and semi-arid climate. Deficiency or extreme changes in surface water resources have resulted in the use of groundwater as an auxiliary source for agricultural water supply in high consumption seasons [1]. Decrease in groundwater levels leads to problems such as wells drying up, river flow reduction, soil and water quality reduction, pumping costs increment, and land subsidence.

The decline in the quality of water and arable soils in the region has led to reduction in growth and production of agricultural products. It is essential to use available water resources, including saline water, rationally and sustainably, while to increase agricultural products. On the other hand, due to the population growth it is necessary to provide food security adequately and to manage water irrigation properly [2–34]. Soil and water salinity management is one of the most important challenges facing the agricultural sector in Iran and in the world [19–26].

Increasing economic profit and agriculture water productivity occurs as a result of choosing the appropriate cultivation pattern and correct management schemes in agriculture and improving irrigation structures [17]. Therefore, using calibrated plant simulation model and linking that to appropriate optimization methods, it would be possible to analyze different management scenarios, and appropriate strategies can also be adopted [15]. On the other hand, water stress due to deficit-irrigation combined with the use of low quality soil and irrigation water limits the growth of crop, and hence accurate predictions of yield in these conditions are important [33].

Accurate estimation of production functions for ministry and policy of agriculture planning [35–42], checking the effects of salinity on product yield and production functions [33, 34, 43], determining and evaluating the production changes and income, optimizing water consumption in agricultural sector [18, 30, 38], quantifying the effects of water scarcity in agriculture [2–33, 37–42], and inspecting the effects of climate change on agricultural production [2, 3, 40–42], are essential input information in economic-hydrological modelling studies [12, 32, 40] worldwide. Therefore, production functions should be determined as quantitative relations between crop yield, salinity and water consumption [30–33]. In general, many researches have been carried out in the conditions of simultaneous salinity and drought to estimate the production function of crops in different regions [1, 21, 33, 37, 38]. However, due to high costs associated with collecting experimental data [42] and inability to use production functions at different times and places [37] the production functions can be estimated according to the effect of various quantitative and qualitative values of water and soil on yield using statistical analysis of the process-based models [3, 4, 11, 15]. The AquaCrop model is one of these models, which requires fewer parameters and input data to simulate crop response to water and salinity stress than other simulation models, and is applicable for most agricultural crops all around the world. This model can estimate crop yield in different climatic conditions and various water stress and salinity [25]. The AquaCrop model has been used in many studies in order to simulate performance under salinity stress conditions, indicating acceptable estimates of crop yield [10, 14, 22, 39].

The simulation models can be linked with optimization approaches on field scale: so they can also be applied to optimize irrigation scheduling, water allocation, and determination of cropping pattern [2–4, 36–42]. Various optimization techniques, such as traditional methods of linear programing (LP), non-linear programing (NLP), dynamic programming (DP), and the artificial intelligence search methods, such as genetic algorithms (GA) and simulated annealing (SA), have been widely used to find the optimal solution of the target problem through solving the objective functions subjected to some constraints [27]. Furthermore,

uncertainties are also employed for optimization issues [4]. CWPFs is one of the uncertainties which itself arises from the changes in meteorological factors. It would therefore be desirable to find a practical and convenient way to integrate the simulation and the optimization models for inspecting an optimal irrigation management strategy on a district scale [11].

Efficient decisions at farm level based on water and soil resources are very important before cropping season. Making right decisions provides a useful tool for farmers to be able to adapt a suitable cultivation and production pattern regarding the status of available water and soil resources [4]. Because of the need to produce more food and to reduce water consumption, economic analysis can be effective in selecting the cultivation pattern and proper allocation of water and soil resources on farms [15]. Therefore, this research is an effective and practical tool for optimizing irrigation allocation areas and cultivation pattern. This model suggested two main advantages distinguishing it from other ones: first, using the model in optimization calculations, CWPFs can be easily prepared for different soil units and various climatic conditions and second this approach can consider important factors in soil salinity changes in the model in addition to differences in crop species. Furthermore, introducing different production functions can estimate the effects of climate uncertainty and thus economic risks on optimization [15].

Qazvin plain is one of the most important agricultural areas in Iran which has encountered water shortage and declined soil quality due to persistent droughts. Considering that, salinity can widely affect crop production. Limited studies on salinity issue have so far been done using the AquaCrop model. In the present study, the production function of the dominant products of the plain under water and salinity stress was investigated. The optimization model was used to achieve the optimal economic benefit in connection with the AquaCrop vegetation model. In previous research, the connection between simulation and optimization models has been less considered: yet, these issues have an effective role in study of managing water resources in the region. The practical results of determining the optimal cultivation pattern and increasing economic profit based on the optimal allocation of water and land give an appropriate solution for agricultural water management in these conditions. Therefore, the summery of our objectives: (1) Use a synthetic plant model like AquaCrop by considering the simultaneous flow of water and salt in the root zone and investigating the effect of it on the yield. (2) Extract crop-water production functions from calibrated and validated AquaCrop model under salinity and drought stress and weather conditions. (3) Develop a nonlinear optimization simulator model under uncertain weather conditions for the optimal allocation of irrigation water and product cultivation in order to maximize economic profit and water use productivity in conditions of restricted water and soil resources.

## Materials and methods

### Study area

The study area with a gross area of over 85,000 ha is located in the southwest of Qazvin, Iran, at latitude $35^{\circ} 30^{'}$ to $36^{\circ} 30^{'}$ and longitude $50^{\circ} 30^{'}$. The climate of the region is semi-arid with an average rainfall of 250 mm and average evapotranspiration of 1330–1587 mm per year. Three main crops of this plain are wheat, barley and maize, which were studied in this study. The surface and subsurface soil texture is mostly in the form of loam and silty loam and the salinity of soil saturated extract is 0–12 ds/m (Table 1). The study area is irrigated under modern irrigation and drainage network.

In fact, the amount of surface water available per hectare in all weather conditions is 4558, 3894, and 2876 cubic meters, respectively. Additional information on the amount of irrigation water is provided in Table 2.

Table 1. Soil physical properties of three major soil types in the case study area.

| Soil type | Soil texture | Depth (cm) | Clay <0.002 (mm) | Silt 0.002–0.05 (mm) | Sand 0.05–2 (mm) | EC (ds/m) | Area (ha) |
|---|---|---|---|---|---|---|---|
| S1 | Silty Loam | 0–55 | 14/7±2 | 44/7±3 | 41/7±3 | 2±2 | 9005 |
| S2 | Silty Loam | 0–90 | 37/6±2 | 32±3 | 30/4±3 | 4±4 | 6263 |
| S3 | Clay loam | 0–60 | 35/6±2 | 34/2±3 | 33/2±3 | 10±2 | 553 |

Table 2. Maximum of total available canal water amount quota (the decrease in percentage of each water supply level was calculated in comparison to present).

| Water Supply level | $Q_s$(10$^6$ m$^3$) | | | Decrease $Q_s$(%) |
|---|---|---|---|---|
| | Wet year | Normal year | Dry year | |
| IR1 | 72 | 62 | 46 | present |
| IR2 | 65 | 55 | 41 | 10 |
| IR3 | 58 | 49 | 36 | 20 |
| IR4 | 50 | 43 | 32 | 30 |
| IR5 | 43 | 37 | 27 | 40 |

## Research procedures

First, using meteorological as well as available field data, AquaCrop model was calibrated and validated based on different irrigation and salinity systems in terms of grain yield (t ha$^{-1}$). Then, using SPI drought index, the years studied (2005–2020) were identified as wet, normal and dry years. A calibrated model was implemented for the produce of the area and production functions based on salinity and irrigation water depth for each climatic condition were obtained. Using production functions with the aim of maximizing the profit and productivity of agricultural water and considering the relevant limitations in GAMs software, a program for optimizing the cultivation pattern and the amount of irrigation water was developed. From the results of the model, the amount of cultivated area of each crop and the amount of irrigation water required in optimal conditions were given. Finally, the amount of income and efficiency of water consumption as well as climate risk under the influence of different salinity scenarios and the amount of water available in the area were simulated using the developed model (Fig 1).

## AquaCrop model

The AquaCrop crop model simulates attainable yields for crops as a function of water consumption under different irrigation regimes. AquaCrop directly links crop yields to water use and estimates biomass production from actual crop transpiration through a normalized water productivity parameter, which is the core of the AquaCrop growth engine. In the continuation of efforts to ensure optimal water consumption for adequate food, the Food and Agriculture Organization of the United Nations (FAO) has upgraded the AquaCrop model from the concept proposed to another concept called normalized productivity of plant water consumption [31]. Ease of use, precision, robustness, low need to input data are the advantages of the new model [31]. Eq 1 suggests the relation between the yield and the amount of water consumption:

$$\left(1 - \frac{Y}{Y_x}\right) = k_y\left(1 - \frac{ET}{ET_x}\right) \tag{1}$$

Where $Y_x$ is the maximum yield (kg ha$^{-1}$), Y is the real yield (kg ha$^{-1}$), $ET_x$ is the maximum

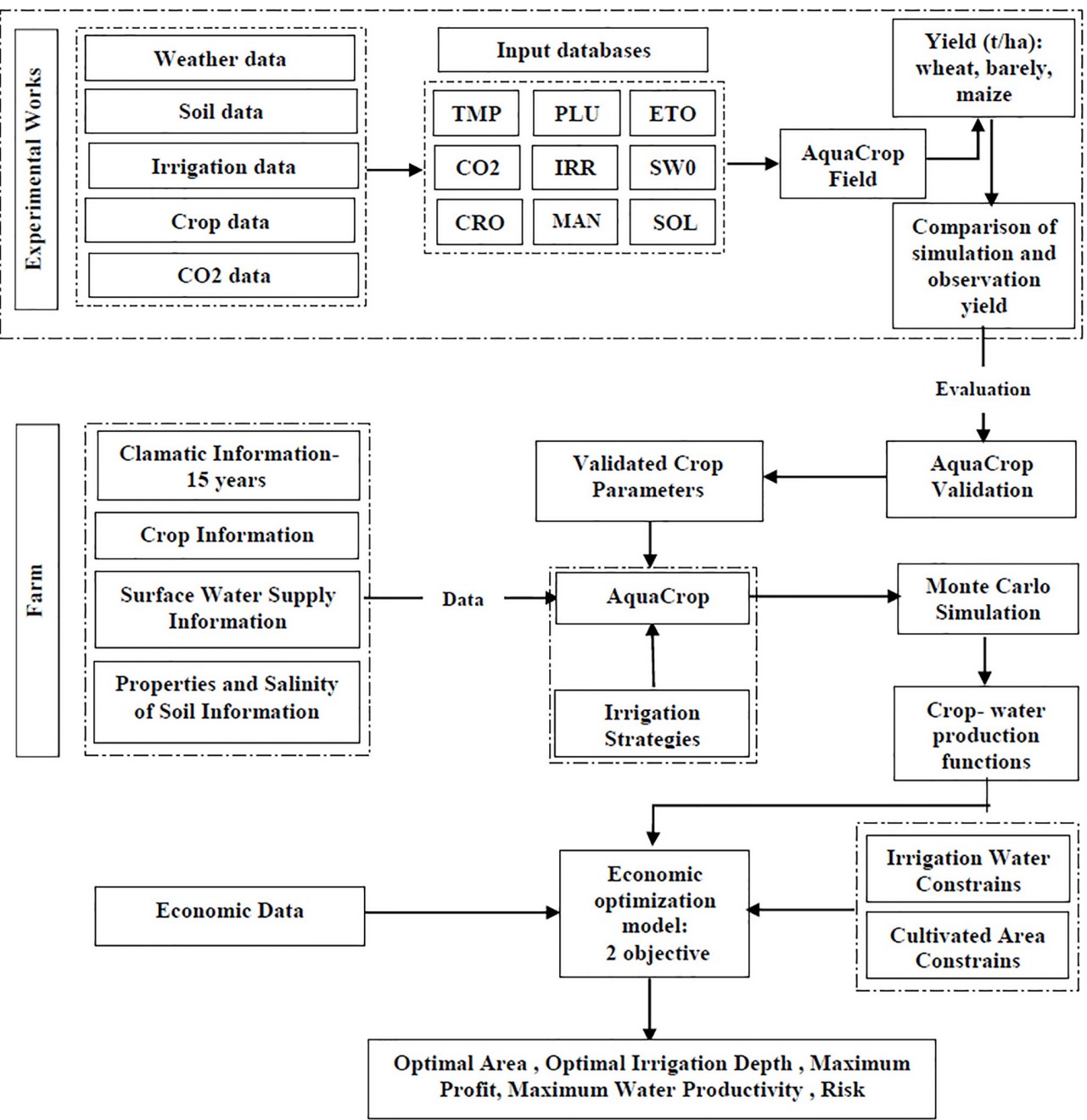

**Fig 1. Flowchart of optimizing the water and land allocation in the region.**

evapotranspiration (mm), and ET is the real evapotranspiration (mm), $k_y$ is the proportion factor between relative reduction in yield and relative decrease in evapotranspiration. The AquaCrop model uses Eq 2 to calculate biomass performance.

$$B_i = w_p^* \times \frac{Tr_i}{Eto_i} \tag{2}$$

In this equation $B_i$ is the biological performance (kg m$^{-2}$), $w_p^*$ (kg m$^{-2}$) is the normalized

water productivity and is a constant parameter, $Tr_i$ is the daily transpiration (mm), and $Eto_i$ is the daily evapotranspiration (mm).

Throughout the crop growth period, the amount of water stored in the root area is simulated through the water balance of inflow (irrigation and rainfall) and outflow (runoff, deep penetration and transpiration evaporation) water in root area. The intensity of water stress coefficients ($K_s$) affecting the canopy cover (CC) of aging, the reduction of canopy cover and harvest index (HI) are determined by the water drain fraction in the root area. Finally, the crop yield is calculated using the mass of aerial part of the vegetation and the adjusted harvest index. AquaCrop divides the soil profile to 12 layers, each layer with 2 to 11 cells to quantify soil salinity. The number of cells depends on soil type of the horizontal layers. Since salt molecules are strongly absorbed by the clay particles, the horizontal clay layers have more cells than sandy ones. The model simulates soil saturated extract, using Eqs 3–5:

$$W_{cell} = 1000 \times \Delta Z \times \frac{\theta_a}{n} \tag{3}$$

$$Salt_{cell} = 0.64 \times W_{cell} \times Ec_{cell} \tag{4}$$

$$EC_e = \frac{\sum_1^n salt_{cellj}}{0.64(1000 \times \theta_{sat} \times \Delta z)} \tag{5}$$

Where $W_{cell}$ is the cell volume (mm), $\theta_a$ is the soil saturation moisture ($m^3 m^{-3}$), $\Delta Z$ is the soil layers thickness (m), n is the number of cells, $salt_{cellj}$ is the amount of salt per cell ($g m^{-2}$), $EC_{cell}$ is the electrical conductivity of a cell (dS $m^{-1}$), $EC_e$ is the electrical conductivity of soil deep saturation extract (dS $m^{-1}$).

## Calibration and validation of AquaCrop

The AquaCrop model must be calibrated and validated for the study area before it is used for the first time. For this purpose, it is necessary to collect field studies on selected products in the desired or adjacent areas. The model parameters related to soil hydraulic properties and crop growth had to be firstly calibrated and validated with the experimental data. This should include different irrigation treatments to allow proper calibration as well as validation. To calibrate AquaCrop for wheat, barley and maize we used the data collected with Golkar et al. [6], Mirlatifi & Sotudenia. [20], Farhadi bansouleh. [41] respectively. that reported by Ramezani et al. [26]. The field experiments for wheat and barley were in Karaj (1998) and for maize in Qazvin (2002). The soil textures were loam with 12, 12, 4 different salinity and irrigation treatments. The irrigation intervals were 7 days. Basin irrigation was used in two experimental places and clay loam with 12 and 7 different salinity and irrigation treatments, respectively. The climate data of corresponding experimental years were available from the Karaj and Qazvin synoptic stations.

To validate AquaCrop for wheat, we used the data reported by Mohammadi et al. [21], in which the field experiments were in Birjand (2006). The soil textures were clay loam with 7 different salinity and irrigation treatments. The irrigation intervals were 10 days, and basin irrigation was used in experimental place. The climate data of corresponding experimental years was available from Birjand meteorological station. To validate AquaCrop for barley, the data reported by Pirasteh Anousheh et al. [24] in Yazd was used. The experiment consisted of two salinity treatments in a sandy loam soil with basin irrigation: irrigation intervals were 7 days. The climate data used for the experimental corresponding years was obtained from Yazd synoptic stations. Data for maize experiment were available from Heidarinia et al. [9] in Ahvaz

**Table 3. Experimental data sets used in calibration and validation of AquaCrop with information: Location, year, date of planting, irrigation and yield in first level of salinity.**

| Crop | year | Location | Soil EC (dS/m) | water EC (dS/m) | n | Date of planting | Yield (t/ha) | Irrigation (mm) |
|------|------|----------|----------------|-----------------|---|------------------|--------------|-----------------|
| Wheat | 2015 | (36°,16´) N | 2 | 1.4–4.5–9.6 | 12 | Nov-14 | 5/6 | 400 |
| | | (59°,38´) E | | | | | | |
| | 1998 | (35°,48´) N | 5.3 | 0.4 | 12 | Nov-6 | 4/4 | 660 |
| | | (51° 0´) E | | | | | | |
| Maize | 2002 | (36°,15´) N | 0.93 | 0.4 | 8 | May-26 | 12 | 763 |
| | | (49°,55´) E | | | | | | |
| | 2017 | (32°,53´) N | 0.5 | 2–4.5–7 | 12 | July-23 | 5/7 | 742 |
| | | (55°, 13´) E | | | | | | |
| Barley | 2017 | (35°, 29´) N | 5 | 2–12 | 4 | Nov -19 | 6 | 319 |
| | | (51°, 40´) E | | | | | | |
| | 1998 | (35°, 48´) N | 5.3 | 0.4 | 11 | Nov -1 | 6/8 | 650 |
| | | (51° 0´) E | | | | | | |

These data were reported in: Mohammadi et al (2016), Golkar et al (1998), for wheat: Mirlatifi and Sotudenia (2002), Heidarinia (2017), for Maize. Farhadi bansule (1998), Pirasteh Anosheh (2017) for Barley. N: number of treatments.

(2016). The experiment was carried out in a deep clay loam soil under 3 salinity and water treatments (irrigation events every 7 days). Basin irrigation method was used in the experiment: All experiments are reported in detail in corresponding papers [9, 21, 24, 26]. Some of the field data obtained from these studies are presented in Table 3 that were used for calibrating and validating of AquaCrop model.

## Model evaluation

For AquaCrop calibration, linear regression between observed and simulated results of the seed yields (t ha$^{-1}$) was drawn, and the correlation coefficient was determined. The statistical characteristics used to compare the simulated results with the actual values include normalized root mean square error (NRMSE), Nash-Sutcliffe coefficient (NSE), and coefficient of determination (R$^2$), which are expressed by Eqs 6 and 7,

$$NRMSE = \frac{1}{O} \times \sqrt{\frac{1}{n} \times \sum_{i=1}^{n} (S_i - O_i)^2} \tag{6}$$

$$NSE = \frac{\sum_{i=1}^{n} (O_{i-}O)^2 - \sum_{i=1}^{n} (S_{i-}O_i)^2}{\sum_{i=1}^{n} (O_{i-}O)^2} \tag{7}$$

Where S$_i$ is the predicted value of yield (t ha$^{-1}$), O$_i$ is the measured value of yield (t ha$^{-1}$), n is the number of observations, and O is the mean measured value of yield (t ha$^{-1}$). The optimum value of the NRMSE in modeling is less than 10%. The NRMSE values between 10 to 20% indicate appropriate status, between 20 to 30% shows average status, and more than 30% indicates invalidity. The NSE coefficient calculates the relative difference between predicted and observed values in the selected statistical period. The value of NSE between 0.75 to 1, 0.65 to 0.75, and 0.5 to 0.65 indicates that the evaluation is very good, good, and satisfactory, respectively. However, if it is less than 0.5, the evaluation is unacceptable. The coefficient of determination R$^2$ is the distribution criterion between the predicted and measured values. If all the predicted and measured values are equal, the value of R$^2$ will be equal to one. The calibrated

**Table 4. Summary of parameters used in AquaCrop model.**

| | Parameter | Unit | Barley | Wheat | Maize | Remark Barley | Remark Wheat | Remark Maize |
|---|---|---|---|---|---|---|---|---|
| 1 | Base temperature | ˚C | 0.00 | 0.00 | 8.00 | D | D | D |
| 2 | Upper temperature (cut-off) | ˚C | 26.00 | 26.00 | 30.00 | C | D | D |
| 3 | Canopy cover per seeding at 90% emergence (CC0) | Cm$^2$/ plant | 1.50 | 1.50 | 6.13 | D | D | C |
| 4 | Canopy growth coefficient (CGC) | %/day | 5.56 | 4.53 | 15.44 | C | C | C |
| 5 | Maximum canopy cover (CCx) in fraction soil cover | %/day | 82.50 | 86.00 | 87.33 | D | D | D |
| 6 | Crop coefficient for transpiration at CC = 100% | - | 0.63 | 1.07 | 1.08 | C | C | C |
| 7 | Canopy decline coefficient (CDC)at senescence | %/day | 6.44 | 9.77 | 11.33 | C | C | C |
| 8 | Water productivity | Gram/m$^2$ | 14.33 | 15.00 | 33.85 | C | D | C |
| 9 | Leaf growth threshold p-upper | - | 0.67 | 0.63 | 0.65 | C | C | C |
| 10 | Leaf growth threshold p- lower | - | 0.22 | 0.22 | 0.14 | C | C | C |
| 11 | Leaf growth stress coefficient curve shape (f shape) | - | 3.00 | 4.93 | 2.90 | C | C | C |
| 12 | Stomatal conductance threshold p-upper | - | 0.62 | 0.61 | 0.69 | C | C | C |
| 13 | Stomata stress coefficient curve shape(fshape) | - | 2.83 | 2.35 | 6.00 | C | C | C |
| 14 | Senescence stress coefficient p-upper | - | 0.73 | 0.69 | 0.67 | C | C | C |
| 15 | Senescence stress coefficient curve shape | - | 3.00 | 2.37 | 2.70 | C | C | C |
| 16 | Reference harvest index | % | 39.25 | 43.02 | 48.00 | D | D | D |
| 17 | ECe threshold p-lower | ds/m | 7.00 | 6.00 | 2.00 | C | D | C |
| 18 | ECe threshold p-upper | ds/m | 20.00 | 19.00 | 9.00 | C | C | C |
| 19 | the form sowing to emergence | day | 20.00 | 12.67 | 5.00 | C | C | C |
| 20 | the form sowing to start of senescence | day | 203.0 | 204.0 | 111.5 | C | C | C |
| 21 | the form sowing to maturity | day | 236.0 | 239.0 | 136.0 | C | C | C |
| 22 | the form sowing to flowering | day | 175.0 | 180.0 | 61.00 | C | C | C |
| 23 | the length of flowering stage | day | 13.00 | 13.00 | 10.00 | C | C | C |
| 24 | time from sowing to maximum rooting depth | day | 98.00 | 94.00 | 104.0 | C | C | C |

(The model was run in the calendar day mode), D: default parameter, C: calibrated parameter.

parameters of AquaCrop model are presented in Table 4. These coefficients in model calibration were based on different references and similar studies on salinity [21].

As described for the AquaCrop model, the various coefficients defined for each crop in the model must be calibrated using field data. In this study, the AquaCrop model was calibrated for three important crops, namely wheat, barley and maize. In this study, deficit-irrigation and salinity were the parameters used to calibrate the model. The statistical criteria after calibration for two series of calibration and validation data for the three crops are reported in Table 5.

The highest value of NRMSE for calibration is 9.14 for Maize and the lowest error is 5.27 for wheat. The value of this statistic was estimated to be less than 10% for all three products, indicating the ability of the model to calculate the products yield. In a similar study, Mohammadi et al. [21] reported NRMSE<10% in the simulation of yield of two types of wheat in salinity and irrigation conditions. Jiang et al. [11] presented the range of yield changes less than 10% for wheat and barley and less than 20% for maize, indicating the ideal condition of wheat and barley and the proper condition of maize. By evaluating the AquaCrop model in estimating the yield of four wheat cultivars and four salinity levels of irrigation water, Kumar et al. [16] obtained the error equal to 1.92–12.76% in predicting seed yield and different salinity levels, so that higher errors were observed in higher salinity levels. Although decreasing trend of crop yield under influence of water stress and salinity is well-simulated by the model, model accuracy in yield simulation under high stress conditions decreases. Modelling

**Table 5. Statistical criteria for calibration of AquaCrop.**

| Crop | Variable | type | Range of observed data (t ha⁻¹) | Range of simulated data (t ha⁻¹) | NSE | NRMSE | R² | Slope | Intercept |
|---|---|---|---|---|---|---|---|---|---|
| **Barley** | Yield (t ha⁻¹) | cal | 2–5.86 | 1.73–5.5 | 0.87 | 7.91 | 85 | 0.8361 | 0.4712 |
| | | val | 3.3–6.2 | 3.8–6 | 0.80 | 8.79 | 81 | 0.7476 | 0.8442 |
| **Wheat** | Yield (t ha⁻¹) | cal | 3.30–6.32 | 2.43–6.75 | 0.90 | 5.27 | 92 | 1.251 | -1.2494 |
| | | val | 2.4–4.4 | 2.6–5 | 0.84 | 8.18 | 89 | 1.2881 | - 1.4549 |
| **Maize** | Yield (t ha⁻¹) | cal | 2.36–11.33 | 1.58–11.56 | 0.97 | 9.14 | 96 | 1.0571 | - 0.1909 |
| | | val | 1.8–5.5 | 2–5.8 | 0.89 | 10.53 | 91 | 1.0065 | 0.0806 |

(Normalized root mean square error -NRMSE-, Nash Sutcliffe coefficient -NSE-) when comparing observed and simulated value of harvestable yield (Y). Slope, intercept, and R2 are for linear regression of observed against simulated values. cal: calibration, val: validation.

efficiency (NSE) was equal to 0.90, 0.87, and 0.98 for wheat, barley, and maize, respectively. As in previous studies, this statistic was estimated more than 90% for the mentioned crops [4, 14, 26]. According to the values of $R^2$ and NSE, the efficiency of the model can be considered acceptable in simulating crops yields in the simultaneous salinity and deficit-irrigation, while barley showed better results under no salinity stress conditions [9, 13, 14, 39]. It appears that since the model only involves the mass transfer and dispersion processes in salinity simulation, the model error increases with increasing salinity [8].

## Crop water production functions (CWPFs)

The simulation results were used as a function of irrigation water after validation. The CWPFs was obtained through fitting the relationship of different levels of total irrigation water amount in the field and the corresponding maximum crop yields for each crop-soil unit [11]. Model simulations were then performed with different levels of irrigation during crop growth period. The irrigation water was initially set to rainfall condition, and then incrementally increased up to the specified maximum value. The simulation avoided water stress in very sensitive stages.

The AquaCrop was run in the calendar day mode. First, AquaCrop model was implemented using the climate data and the information on the dominant crops (wheat, barley, maize) in the region, as well as information on irrigation water supply from a surface water source. The climate information required by AquaCrop includes minimum temperature, maximum temperature, rain, and ET0.

The reference evapotranspiration was calculated using the FAO Penman Monteith equation. It was derived from the daily climate data (mean temperature, radiation, relative humidity, and wind speed). A 15-year data set (2005–2020) was available from the Qazvin synoptic station (36˚15´N, 50˚03´E). Basin irrigation method was used. According to the available information, the irrigation efficiency of the region was considered to be 35% [29]. The planting date of the crops and the soil properties are presented in Tables 1 and 3, respectively. The CWPFs are also affected by different climate conditions in different years. Therefore, many years of simulations need to be conducted to determine the functions that consider climatic effects and uncertainties on crop yields.

In the studied 15 years, according to the SPI index, the average wet, normal, and dry years were considered as wet, normal and dry weather conditions, respectively. The effects of climate uncertainty can affect optimization. Random numbers were generated based on deficit-irrigation in different months and their corresponding yields based on the Monte Carlo method. The Crop yield was calculated for different amounts of irrigation water and salinity in each area. The water stress conditions during growth period related to each plant were considered. Then, using the non-linear regression model, the best fit between points was introduced as a

performance function. For each crop-soil unit, representing wet, average and dry climate conditions Thus, the effects of climate uncertainties could be included in optimization.

## Optimization model

With the help of performance function extracted from the model under salinity and irrigation stresses at different levels, by developing an economic optimization model, an optimal management of irrigation water and the area under cultivation was achieved. By combining the GAMS (The general algebraic modeling system) model and the output functions of AquaCrop model, constraints include the amount of irrigation water consumption and the area under cultivation, the objective functions for maximizing economic profit and the water productivity were obtained (Fig 1). An economic optimization model was developed to maximize the net profit and water consumption efficiency (Eqs 8 and 9) in the region, considering the water constraints and cultivation levels (Eqs 10 and 11). According to the performance functions for irrigation water, a non-linear planning model was defined. In this research, to solve the optimization problem, a classical two-objective optimization method was utilized using the limited epsilon method of the GAMs software [3]. The economic information is given in Table 6.

$$OF1:\ Max(TGM) = \sum_{S=1}^{3}\sum_{j=1}^{3}\left\{a_{js} \times \left[p_j \times O_{js} - C_j - C_w \times W_{js} \times 10^{-5}\right]\right\} \quad (8)$$

$$OF2:\ Max(WP) = \sum_{s=1}^{3}\sum_{j=1}^{3}\left\{O_{js}/(W_{js} \times 10)\right\} \quad (9)$$

$$\sum_{s=1}^{3}\sum_{j=1}^{3} a_{js} \leq A \quad (10)$$

$$\sum_{s=1}^{3}\sum_{j=1}^{3} W_{js} \leq TWS \quad (11)$$

where TGM is the profit ($) from all the crops cultivated in the farms: $p_j$ is the price ($ kg$^{-1}$) of crop j: $a_{js}$ is the areas (ha) devoted to crop j in farm s: $O_{js}$ is the crop yield (kg ha$^{-1}$) as a function of seasonal irrigation $w_{js}$ (mm) of crop j in farm s: $C_j$ is the fixed production costs per unit area ($ ha$^{-1}$) for crop j: $C_w$ is the water allocation cost per unit volume ($ m$^{-3}$). TWS is the total amount of water demanded for the region (m$^3$), A is the irrigable area of the farms (ha).

**Table 6. Details of existing crops and economic information in the normal year.**

| Soil type | Crops | Cultivation area | | Maximum yield | Water requirement | Benefit |
|---|---|---|---|---|---|---|
| | | ha | % | (t ha$^{-1}$) | (mcm) | ($ kg$^{-1}$) |
| S1 | Wheat | 7663 | 85 | 6.24 | 22.9 | 0.14 |
| | Barley | 1169 | 13 | 5.24 | 14.3 | 0.11 |
| | Maize | 173 | 2 | 9.24 | 3.5 | 0.09 |
| S2 | Wheat | 5449 | 87 | 6.03 | 9.3 | 0.14 |
| | Barley | 626 | 10 | 5.12 | 0.95 | 0.11 |
| | Maize | 188 | 3 | 7.52 | 1.14 | 0.09 |
| S3 | Wheat | 514 | 93 | 5.02 | 0.93 | 0.14 |
| | Barley | 28 | 5 | 4.36 | 0.4 | 0.11 |
| | Maize | 11 | 2 | 4.86 | 0.5 | 0.09 |

Next to solve proposed two-objective programming model we employed the 'ε-constraint method':

$$min\left\{f_1(x).f_2(x).\ \ldots.f_p(x)\right\}$$
$$s.t. \qquad x \in S$$

(12)

where x is the vector of decision variables, $f_i(x)$ is the ith objective function, p is the number of objective functions, and S is the feasible region. A feasible solution x is said to be efficient and the corresponding objective function is said to be non-dominated if there is no other feasible solution x such that fi(x′) ≤ fi(x) for every i = 1, 2,..., p, with at least one strict inequality.

In the ε-constraint method, one of the objective functions is optimized, and other objective functions are incorporated as constraints into the constraint part of model as follows:

$$min \qquad f_1(x)$$
$$s.t. \qquad f_2(x) \le \varepsilon_2$$
$$f_3(x) \le \varepsilon_3$$
$$\ldots$$
$$f_p(x) \le \varepsilon_p$$
$$x \in S$$

(13)

where εi is the satisfaction level of objective function i and solutions can be obtained by parametric variations of the satisfaction levels ε2, ε3,..., εp, on the right-hand side of the constrained objective functions. If some of the objective functions become maximized, the related constraint should be in the form of fi(x) ≥ εi. The steps of ε-constraint method can be summarized as follows: Step 1: Solve p − 1 single-objective problems (SOPs) and find the optimum solution and related objective function value SOPi:

$$optimize\,f_i(x) = x_i^*.f\left(x_i^*\right)$$
$$i = 2.\ldots.p$$
$$s.t. \qquad x \in S$$

(14)

Step 2: Using the solution that optimizes the ith objective function, calculate values for other objective functions. These values form the ith row of the payoff table. Applying this approach, all of the rows of payoff table are determined. For each column i, determine the minimum and maximum values of objective function i. The structure of payoff table is illustrated (Fig 2).

Step 3: Vary the value of ε in the range of corresponding objective functions resulting from payoff table, that is, yimin ≤ εi ≤ yimax. Often the range of objective functions is segmented into equal parts, and the grid points are used as the values of ε. Step 4: If the decision maker is satisfied with one of the generated solutions, stop and select the preferred solution as final decision: otherwise, select the most preferred segment and vary the value of ε in the new range to generate new pareto-optimal solution.

Optimization was conducted for each water supply level then optimal irrigation allocation among different canals and crop-soil units could be obtained, and the corresponding planting area was also optimized. In order to consider the uncertainties of CWPFs in different climate conditions, the optimization was conducted for each level of water supply using the three types of CWPFs curves (normal, dry and wet). Other factors including irrigation method, total crop area, crop variety, canal water conveyance efficiency, etc., were assumed to be unchanged.

| $SOP_i$ | $f_2(x_i)$ | $\cdots$ | $f_p(x_i)$ |
|---|---|---|---|
| $x_2^*$ | $f_2(x_2^*)$ | $\cdots$ | $f_p(x_2^*)$ |
| $\vdots$ | $\vdots$ | $\cdots$ | $\vdots$ |
| $x_p^*$ | $f_p(x_p^*)$ | $\cdots$ | $f_p(x_p^*)$ |
| $y_i^{min}$ | $f_2(x_2^*)$ | $\cdots$ | $f_p(x_p^*)$ |
| $y_i^{max}$ | $max\,(f_2(x_i))$ | $\cdots$ | $max\,(f_p(x_p))$ |

**Fig 2. Structure of payoff table in the ε-constraint method.**

When using one type of CWPFs, the optimal strategy may have an economic risk caused by possible yield reduction due to annual climate variation. In order to assess the economic risk, a new variable was introduced and defined as follows [4]:

$$R = MAX\left\{|P - P_f|.|P - P_u|\right\} \qquad (15)$$

where $P_f$ and $P_u$ are the total benefits for the wet and dry years, respectively.

## Results

### Crop water production functions

For any economic analysis of irrigation water quantities, the production functions give a mathematical relation between crop yield and irrigation water. The yield function (yield-seasonal irrigation) in each irrigation season (for three years of wet, normal, dry year) for each of the 9 crop-soil units (three soil types, for three crops of wheat, barley, and maize) was obtained using the AquaCrop model (Fig 3). For example, under normal climatic conditions, the yield increases linearly for each unit of soil and crop. Maximum yields for barley, wheat, and maize occur at the irrigation depths of 300, 400, and 1000 mm, respectively, and is fixed at greater depths. Similar to previous studies, quadratic functions were used to quantify the non-linear relation between the crop yield and the amount of irrigation water [21, 30, 36, 37]. In rained conditions for wheat and barley, as salinity increases, the yield decreases. This behavior is more evident in the salinity of 12 ds/m. This is quite consistent with the properties of soil and the water capacity of soil as well as increase in the osmotic potential (Fig 3). As expected, the yield is higher in type I soils with lower salinity. The yield reduction in different soils under the same climatic conditions is 10 to 50% for maize, 2 to 40% for wheat, and 2 to 30% for barley.

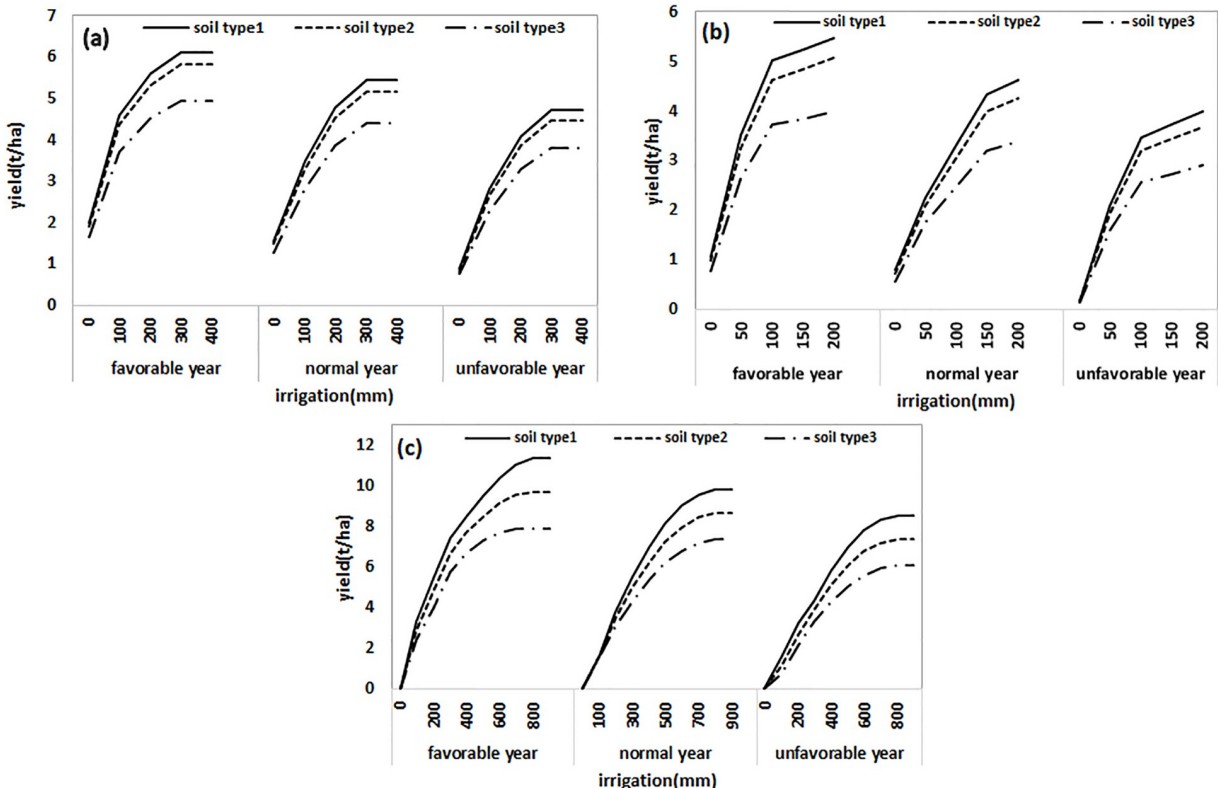

**Fig 3. Crop yield function in different soils (soil type1: EC<4, soil type2: 4<EC<8, soil type3: EC>8 ds m$^{-1}$) under different climatic conditions (wet, normal, dry year): (a) wheat, (b) barley, and (c) maize.**

This can be explained by the increase in salinity and changes in soil texture. The coarser soil texture, the more water can be stored in water deficit conditions [11]. With increasing salinity under the same irrigation conditions, the yield decreases due to the sensitivity of crops to salinity. According to studies, in terms of sensitivity to salinity, maize is more sensitive than wheat, and wheat is more sensitive than barley. In the more irrigations the yield in saline soils is reduced by half in non-saline soils. In saline conditions, more irrigation water is needed. Simultaneous salinity and deficit irrigation have a greater effect on the yield. On the other hand, the yields for the same irrigation in the same soil are not similar in different years, owing to the differences in rainfall, temperature, and wind speed [11]. The maximum reduction in the yield of maize, barley and wheat in various climatic conditions was 60, 30 and 30%, respectively.

As expected and can be seen in Fig 3, the production functions of CWPFs varies in different soils with different salinities, and in various climatic conditions. In soils with higher salinity, water stress and salinity with adverse effects on soil water potential energy and increase in osmotic pressure, cause low crop growth, reduced water uptake by crops, and consequently, reduced yield [7]. In general, the higher the salinity is, the lower the yield will be. Jiang et al. [11] considered soil type to be effective in the relation between yield and amount of irrigation water. However, salinity is not the only factor effective in reducing yield, and factors such as climate, soil type and electrical conductivity are also determinant. Kim et al. [15] in addition to salinity, introduced soil saturated hydraulic conductivity as an effective factor.

**Table 7. Optimized maximum total benefit the average net irrigation depth and planting area proportion for each crop in the irrigation district.**

| Water Supply level | Total benefit (USD) | Economic risk ($10^6$ USD) | Range of irrigation(mm) | | | Range of planting area Proportion (%) | | |
|---|---|---|---|---|---|---|---|---|
| | | | Wheat | Barley | Maize | Wheat | Barley | Maize |
| present | [4.88,2.27] | 1.59 | [392,438] | [255,425] | [909,1044] | [85,91] | [9,13] | [2,3.8] |
| IR1 | [5.85,2.61] | 1.94 | [371,413] | [260,365] | [859,944] | [69,77] | [22,28] | [0.1,2.4] |
| IR2 | [4.99,2.53] | 1.82 | [318,372] | [232,324] | [764,871] | [69,78] | [21,27] | [0.1,2.4] |
| IR3 | [4.54,2.25] | 1.70 | [278,330] | [206,284] | [668,855] | [70,86] | [14,26] | [0.1,2.3] |
| IR4 | [3.84,1.78] | 2.08 | [239,289] | [181,244] | [573,795] | [77,86] | [13,21] | [0.1,1.2] |
| IR5 | [3.13,1.14] | 2.39 | [199,248] | [155,207] | [476,734] | [76,86] | [13,22] | [0.0,1.2] |

(IR1: full irrigation, IR2: 10% deficit-irrigation, IR3: 20% deficit-irrigation, IR4: 30% deficit-irrigation IR5: 40% deficit-irrigation)

## Optimized results

Using the CWPFs and the optimization model, the optimized irrigation water allocation and cultivation area were obtained at different water supply levels for the yield. The optimized total benefit (P) and economic risk (R) are presented in Table 7. The results showed that the profit could be effectively increased through the optimization of irrigation water allocation and cropping pattern.

The amount of optimized irrigation water allocation for each soil has decreased compared to the present condition. The range of changes in full irrigation conditions is 363–389 for soil type 1, 344–403 for soil type 2, and 339–405 for soil type 3. These changes in the present conditions were estimated to range from 384 to 432, from 398 to 443, and from 410 to 450, respectively. Improper allocation of surface water results in uncontrolled abstraction of groundwater resources and serious environmental challenges e.g., degradation of lacustrine ecosystems, demise of large tracts of vegetation, and land desertification due to less water allocation [42]. So water irrigation allocated is planned to be reduced.

Under the present irrigation water supply, the optimized total benefit reached 5.85 million dollars on maximum and 2.27 million dollars on minimum that an increase of 16–19% compared to the present condition with an economic risk (R) of 1.94 million dollars (Table 7). Due to type of crop and the differences of soil type among different canal command areas, there were clear spatial differences in the allocation of irrigation water. The variations of allocated irrigation water shown in Fig 4 and the optimized average irrigation depth for each crop and the corresponding crop area are provided at different water supply levels are presented in Table 7. The average irrigation amount reduced from 145 to 590 mm for level 0–5 (IR1-IR5), and the allocated irrigation water among different soils was different (Fig 4). In the present situation, with the amount of available irrigation water, the results showed that the amount of irrigation water for three crops have decreased. The area and amount of irrigation water for wheat are 69–77% of the total planting area and 371–413 mm, respectively. The area and irrigation water for maize is 2–3.8% and 859–944 mm in different climatic years, and for barley are 22–28% and 260–365 mm, respectively (Table 7).

It seems that the irrigation water has been first allocated to wheat from 5 water supply level, which is less than the maximum value of 438 mm at water supply levels. The irrigation for the other two crops was also gradually reduced from full irrigation to 40% deficit-irrigation (Table 7) and the box difference became smaller at the lower levels of water availability because wheat and barley were irrigated at their minimum amount of demand. For maize decreased irrigation ranging from 917 to 490 mm for different climatic years. The amount of average

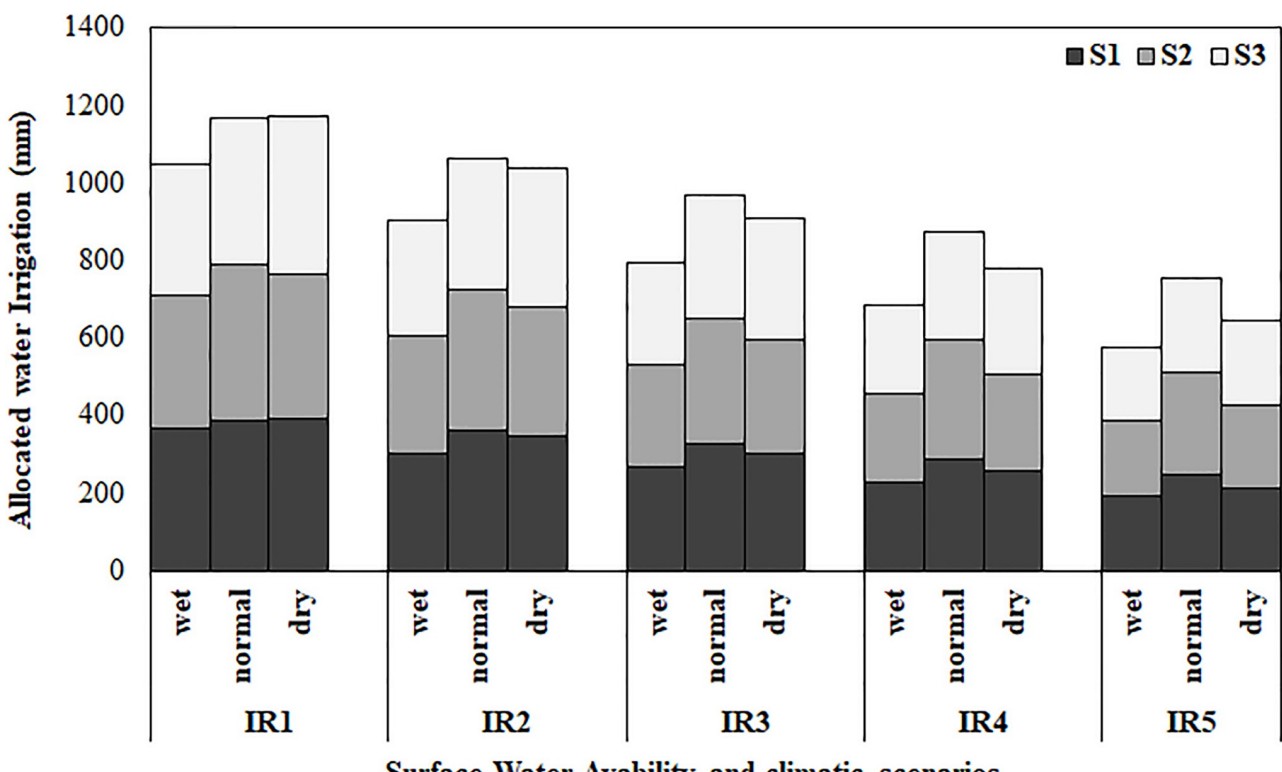

**Fig 4. Optimal allocation of irrigation water amount for different soil (S1: Soil type1 (EC<4), S2: Soil type 2 (4<EC<8), S3: Soil type 3(EC>8)) at different levels of water availability (IR1: No water stress, IR2: 10% water stress, IR3: 20% water stress, IR4: 30% water stress, IR5: 40% water stress) under different climatic conditions (wet, normal, dry year).**

irrigating for these wheat and barley was 260–155 mm and 371–199 mm, respectively form level IR1 to IR5 (Fig 5).

## Allocation of water and land

Regarding the needs of region and the existing cultivation pattern and the potential for their change in the region, the limitations of cultivation area were defined. These limitations were based on the existing cultivation pattern and established in order to increase two-objective functions of increasing profits and increasing the efficiency of agricultural water consumption. It is assumed that all the lands in each sub-network have the same amount of water and the profit of crops can be maximized with proper allocation of water and land.

Fig 6 shows that the lower sensitivity of wheat to water and salinity has increased its attractiveness for most fields. Allocating the majority of a land to wheat will result in revenue and profit increase. Even if the amount of water is low or the area under cultivation is reduced, the amount of wheat planting in saltier soils should be increased. However, the amount of wheat cultivation has decreased compared to the current situation. Due to the special place of in the food basket of Iranians and the policy of the ministry of agriculture jihad in this region, wheat cultivation is the main priority. This issue was considered in the limitations. Yet, in recent years, along with rising livestock prices and the need for more production, the need for forage crops has also increased. As shown in Table 7, by raising the cultivation area of products such as barley, with aim of providing bread and animal feed, the profitability can be increased in the province. Therefore, regarding the economic value of major crops, wheat and barley are the

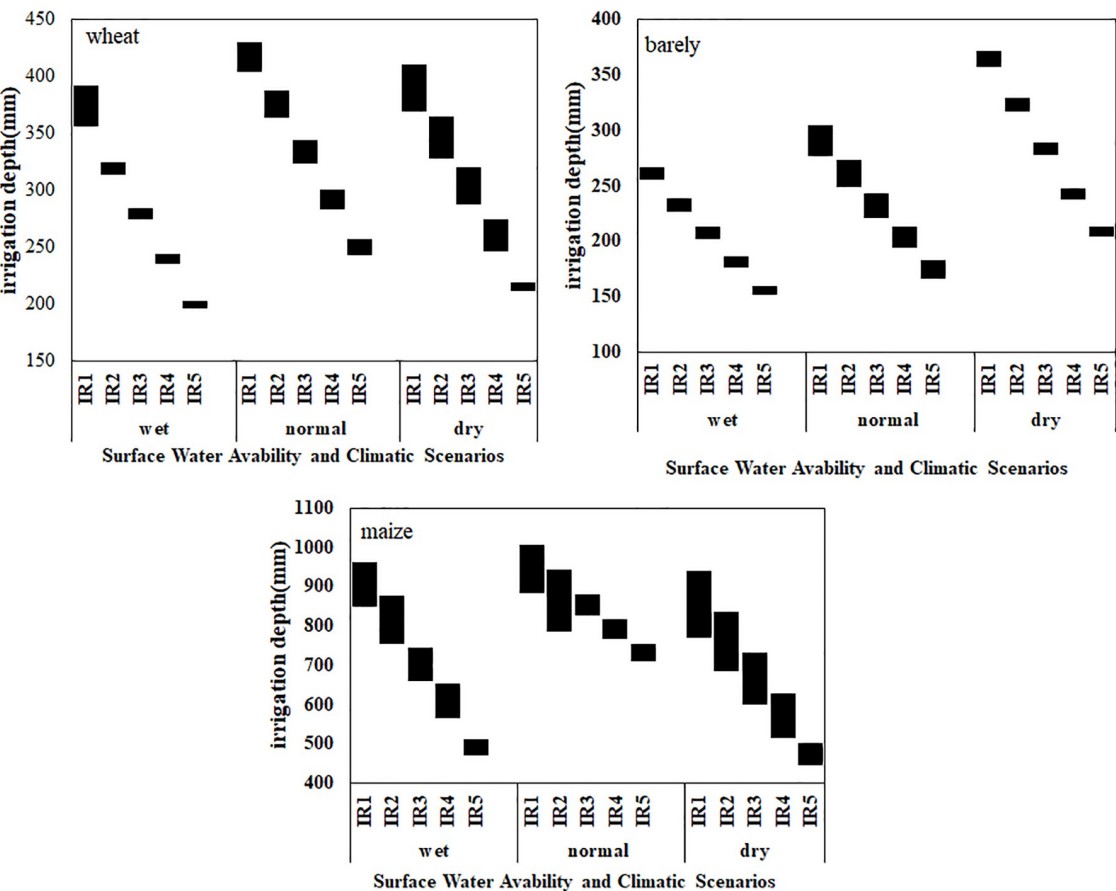

**Fig 5. Optimal total irrigation depth for each crop at different surface water availability levels (IR1: No water stress, IR2: 10% water stress, IR3: 20% water stress, IR4: 30% water stress, IR5: 40% water stress) under different climatic conditions.**

two suggested crops for cultivation in the study area. Because of sensitivity to salinity, maize is not cultivated in very saline soils: in soil with salinity of 4–8 ds/m, only 2% of cultivation is done.

The area of crops as compared to present situation for wheat and maize decreased 18–21% and up to 92%, in soil type1, 15–18%, 28–96% in soil type2 and 11–20%, up to 100% in soil type 2. Whereas the area of barely increased 116–200%, 16–160%, 76–208% respectively in soil type1, 2, 3. The optimal planting area for wheat and maize there were an increase of 33–39% and a decrease 5–10% for level 2–5 (IR2 to IR5) as compared to level 1(IR1) in soil type 2, 3 under different climate years respectively.

## Net profit and water productivity

The product price and producing cost are two important factors in optimal water and land allocation. In this study, an attempt was made to optimize the levels and amount of irrigation water with the two objectives of increasing water productivity and increasing profit is shown in Fig 8.

As expected, in the case of reduced irrigation water and in all three soil types, the profit decreased. As the soil salinity increased, the rate of reduction in profit intensified. In very saline soils, this reduction was partial and negligible up to 30% deficit-irrigation in wet weather conditions and up to 20% deficit-irrigation in dry weather conditions. In non-saline and saline

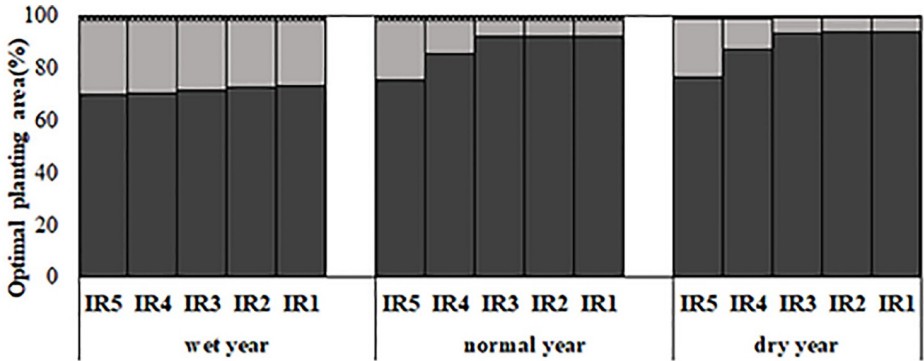

(a)

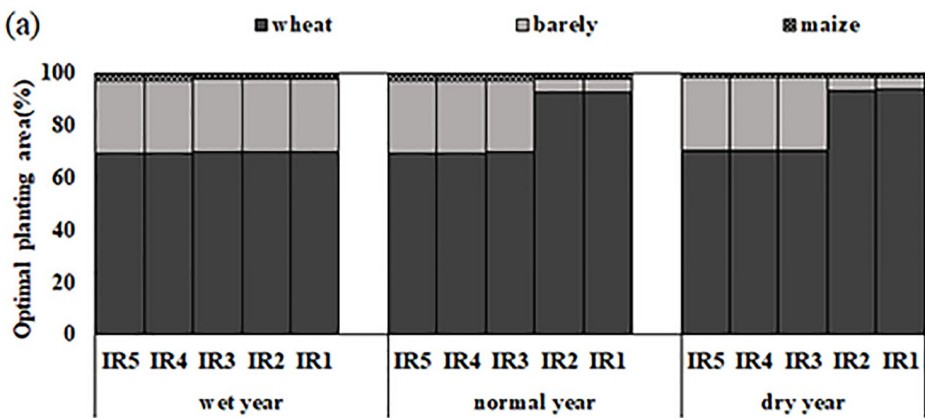

(b)

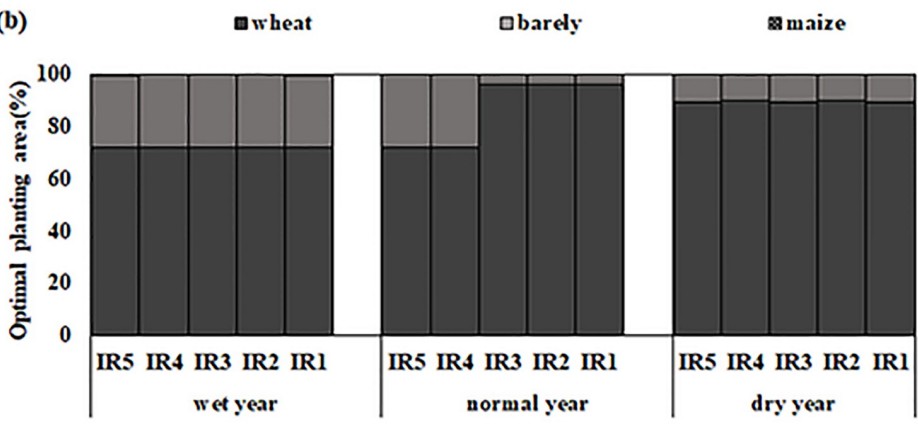

(c)

**Fig 6. Optimal area water for each crop and optimal applied irrigation with different water allocations level (from full irrigation to 40% deficit irrigation) under the different climatic scenarios:(Present: Cultivated area and irrigation demand in present situation), a: Soil type1 (EC<4), b: Soil type2 (4<EC<8), c: Soil type3 (EC>8 ds/m).**

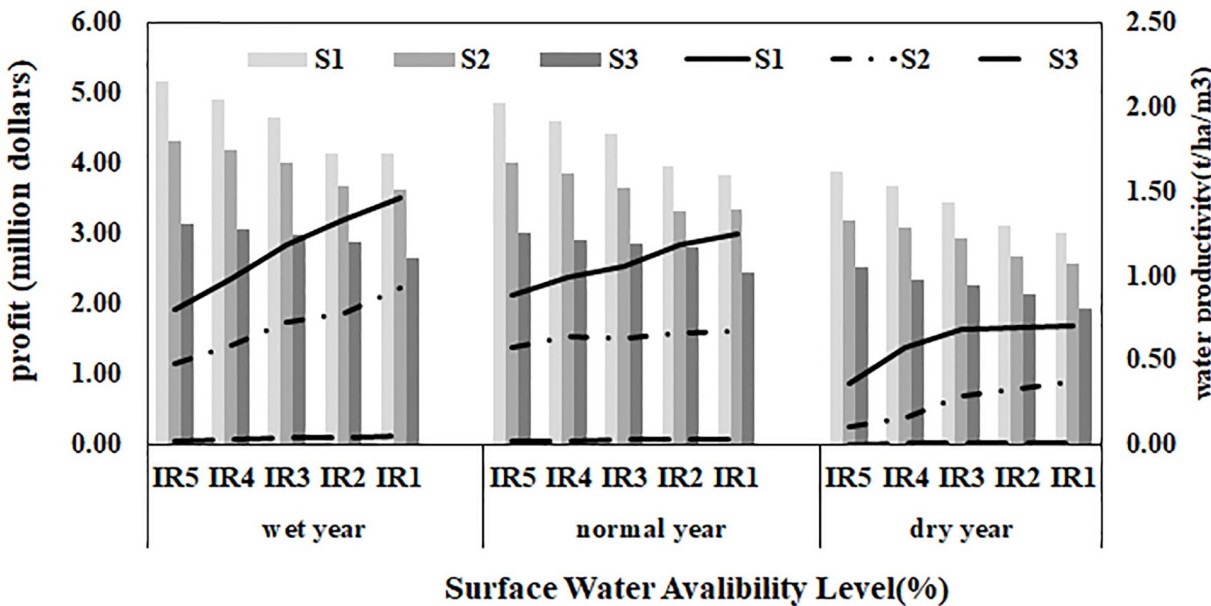

**Fig 7. Profits and water productivity in wet, normal and dry years for soil types: S1, S2, S3.**

soils, a sharp reduction in profit occurred in the >30% deficit-irrigation scenarios. The results of the present study, like previous studies, indicate that the amount of net profit in saline soils is affected by salinity; the higher soil salinity is, the lower the final profit will be [11–14] That also reduces water productivity. Tafteh et al. [29] also showed that water productivity increases by applying deficit-irrigation. The increase in water productivity in all three soil types due to deficit-irrigation is illustrated in Fig 7. According to Fig 8, considering reasonable variation in economic profit as compared in current situation, the amount of profit at 20% of irrigation water stress were seen. At these levels, the agricultural water productivity will be 20, 25, 31% higher than those of the full irrigation, respectively.

## Discussion

Combining the simulation crop model (AquaCrop) with an economic model was an effective tool for optimal allocation of the amount of irrigation water and crop area in each type of soil in Qazvin irrigation network. This model was adopted to generate the crop water production functions (CWPFs) for different soil-crop units while taking climate conditions into consideration. The model could better consider the effects of factors of spatial soil, crop species. In addition, the introduction of three type CWPFs can also include impacts of climate uncertainty into the objective functions, and the economic risks could be provided in optimization. Therefore, model could be more reasonable with considering above mentioned as compared with the previous regional optimization models [28, 29].

Optimized results showed that an increase of 16–20% of the total profit could be obtained and saving of 20% irrigation water (IR4) with the same in present profit for region with an economic risk of 1.07 million dollars under the present water supply level, when taking into account climate uncertainty and the greater risk of climate variations were seen at the lower water supply conditions. Jiang et al. [11] in a research with 23% deficit-irrigation achieved the minimum reduction in profit. Nazr et al. [23] showed that by changing the cultivation pattern,

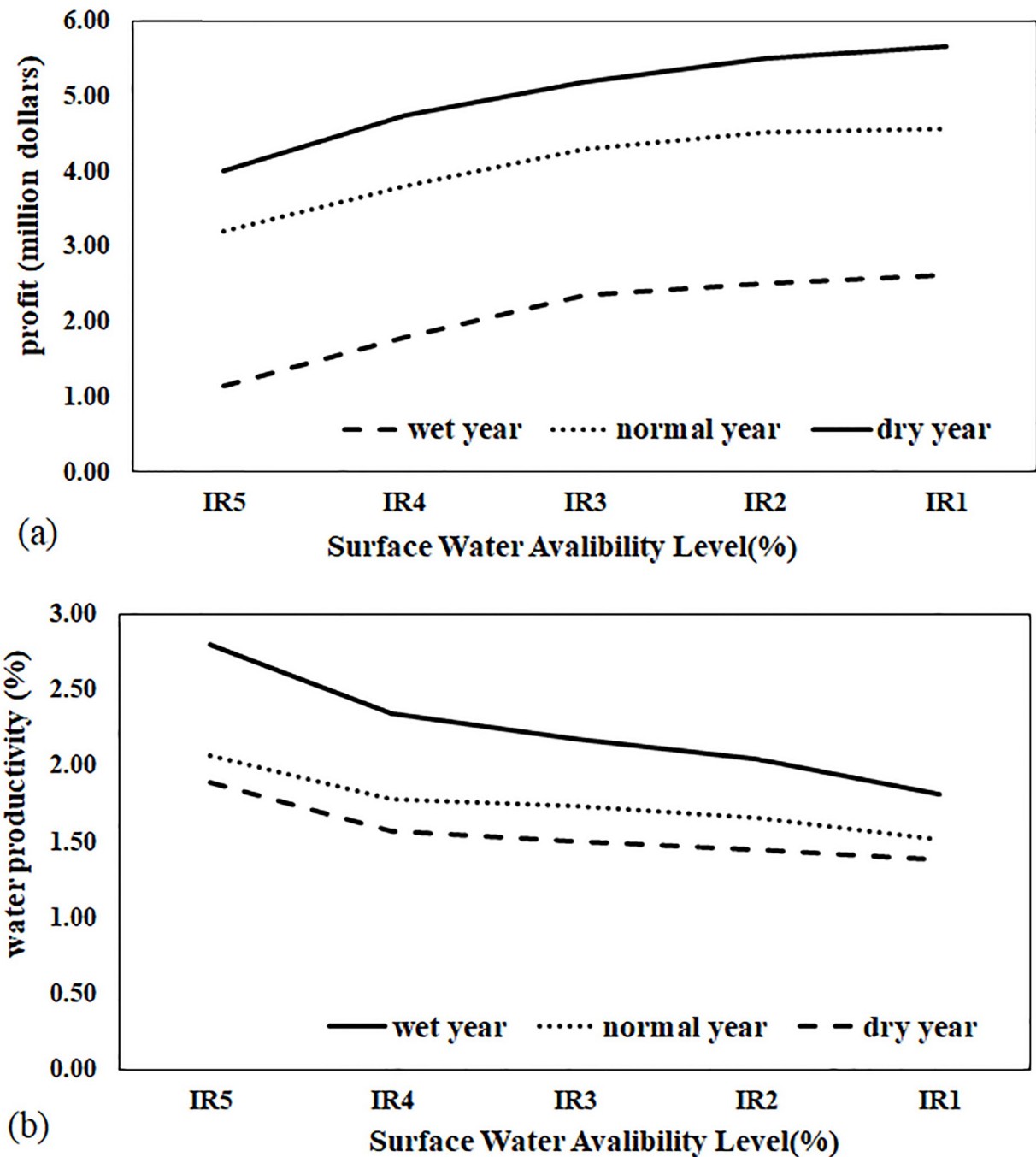

**Fig 8. (a) profits and (b) water productivity in wet, normal and dry years.**

under water and land restrictions, irrigation water decreases by 10%. Shirshahi et al. [28] estimated the volume of optimal water consumption to be about 10% less than full irrigation. The average irrigation amount was reduced from 1256 to 1109 mm for optimal condition as compared to present irrigation that decrease 12% and for level IR1- IR5, decreased from 1109 to 685 mm for different climatic years.

## Conclusion

In this study, a regional economic optimization model was developed based on the production function, extracted from the AquaCrop model, to improve the cultivation pattern and the amount of irrigation water for different soil units and major crops in various climatic conditions. This model includes the production function in soils of varied salinities. Therefore, the model is more comprehensive than those used in the previous optimization studies [4, 5, 15]. In this model, the two-objective functions of profit maximization and water consumption efficiency were used for optimization. In this study, the AquaCrop model was calibrated using field data of wheat, barley, and maize in different treatments of salinity stress and irrigation. The low errors observed between the simulated and actual data denoted that the AquaCrop model was well calibrated for salinity stress and deficit-irrigation. Wheat showed the best response to water and salinity stresses at different levels, and among the studied crops, the highest error belonged to maize. Also, the results of calculations in a part of the Qazvin plain indicated that in the same profit conditions, regarding the current situation, the volume of irrigation water used in all three types of soil and three climatic conditions 20% decreased. Therefore, current situation is not optimal and can be improved by increasing 16–19% in profit and with increasing 12–17% in water productivity in the region. Decision makers and water authorities can use it as an effective tool for such large and complex irrigation planning problems optimizing regional water consumption as well as cultivation patterns in different climatic and soil salinity conditions.

## Supporting information

**S1 Data. The original data and more information related to the manuscript are included in the data.**
(XLSX)

## Author Contributions

**Conceptualization:** Sara Bulukazari, Seyed-Habib Mousavi-Jahromi.

**Data curation:** Sara Bulukazari, Niazali Ebrahimipak, Hadi Ramezani Etedali.

**Investigation:** Niazali Ebrahimipak.

**Methodology:** Hossein Babazadeh, Niazali Ebrahimipak.

**Software:** Hadi Ramezani Etedali.

**Supervision:** Hossein Babazadeh.

**Writing – original draft:** Sara Bulukazari.

**Writing – review & editing:** Hossein Babazadeh, Seyed-Habib Mousavi-Jahromi.

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
